# Semantically-Guided Inference for Conditional Diffusion Models: Enhancing Covariate Consistency in Time Series Forecasting

## Abstract

Diffusion models have demonstrated strong performance in time series forecasting, yet often suffer from semantic misalignment between generated trajectories and conditioning covariates, especially under complex or multimodal conditions. To address this issue, we propose SemGuide, a plug-and-play, inference-time method that enhances covariate consistency in conditional diffusion models. Our approach introduces a scoring network to assess the semantic alignment between intermediate diffusion states and future covariates. These scores serve as proxy likelihoods in a stepwise importance reweighting procedure, which progressively adjusts the sampling path without altering the original training process. The method is model-agnostic and compatible with any conditional diffusion framework. Experiments on real-world forecasting tasks show consistent gains in both predictive accuracy and covariate alignment, with especially strong performance under complex conditioning scenarios.

## 1 Introduction

Time series forecasting is a fundamental problem in a wide range of real-world applications, including energy demand prediction Challu et al. (2023); Somu et al. (2021), healthcare monitoring Morid et al. (2023); Zhou et al. (2024b), and traffic flow analysisMa et al. (2021); Wang et al. (2022). Recently, diffusion models have emerged as a powerful generative framework for probabilistic forecasting, owing to their ability to model complex distributions and capture uncertainty through iterative denoising processesYang et al. (2023); Croitoru et al. (2023); Ho et al. (2020). Several diffusion-based approaches—such as CSDITashiro et al. (2021), SSSDLopez Alcaraz & Strodthoff (2023), TMDMLi et al., and MTSCIZhou et al. (2024a)—have demonstrated promising results across various forecasting tasks.

A major advantage of conditional diffusion models lies in their flexibility to incorporate future-known covariates—such as control signals, policy indicators, or external features—into the generation processRasul et al. (2021). Ideally, the generated forecast should be semantically aligned with these covariates, reflecting plausible future trajectories under the specified conditions. However, in practice, existing methods often fail to ensure such alignment. prior work in diffusion guidance shows that additional guidance signals are frequently required to enforce conditional fidelity Dhariwal & Nichol (2021); Ho & Salimans (2021). The learned conditional distribution $q(x|y)$, obtained through standard denoising training objectives, can deviate significantly from the true posterior $p(x|y)$, especially in scenarios where the covariate space is high-dimensional, multimodal, or exhibits complex dependenciesleepriorgrad, liangtheory.

This misalignment manifests as a lack of consistency between the generated forecasts and the given covariates. While the model may produce diverse and realistic samples, these samples often disregard the semantics implied by the conditioning inputs, leading to unreliable or incoherent predictions. To mitigate sample variability and improve robustness, a common practice during inference is to generate multiple trajectories (e.g., 100 samples) and take their pointwise median as the final forecastLiu et al. (2024); Tashiro et al. (2021). While this heuristic can reduce variance, it does not address the underlying problem of conditional inconsistency. In fact, median aggregation may obscure semantic mismatches by averaging over samples that are individually inconsistent with the

covariates, thereby concealing rather than correcting the model's failure to respond to conditioning signals.

Fundamentally, current diffusion-based forecasters lack mechanisms to evaluate or enforce consistency between intermediate sampling states and covariates during generation. The sampling process is entirely driven by score-based denoising, without feedback on whether the evolving trajectories remain aligned with the conditioning inputs. This highlights a critical gap: the absence of an inference-time corrective mechanism that can dynamically guide the sampling process toward covariate-consistent forecasts.

To address this challenge, we propose SemGuide, a novel inference-time framework that enhances semantic consistency in conditional diffusion models through a lightweight and model-agnostic sampling refinement mechanism. Our key idea is to introduce an external scoring network that assesses the semantic alignment between intermediate sampling states and the given covariates. This scoring function provides a proxy likelihood signal that quantifies how well a partially generated sample agrees with the conditioning input.

During the sampling process, we employ a stepwise importance reweighting strategy, where multiple particles at each denoising step are weighted according to their semantic consistency scores. These weights guide the sampling trajectory toward more covariate-consistent regions of the sample space. The procedure effectively implements a nonparametric posterior correction over the sampling path, without altering the original training objective or architecture of the diffusion model. This results in a self-corrective generation process that dynamically steers sampling toward semantically plausible outcomes under the specified covariates.

Importantly, the scoring network is trained separately using positive and negative pairs of covariates and future sequences from training data, optionally with noise injection to simulate diffusion states. It requires no access to ground-truth labels at test time, and imposes no changes to the base model during training. Thus, our method is compatible with any conditional diffusion model and can be seamlessly integrated into existing pipelines.

This work makes the following key contributions:

- We identify and address a core limitation in conditional diffusion models—semantic misalignment between generated trajectories and conditioning covariates—and demonstrate how this affects forecast reliability in time series applications.

- We propose SemGuide, a model-agnostic, inference-time framework that refines the sampling process via semantic scoring and stepwise importance reweighting, improving consistency without modifying the base diffusion model or training procedure.

- We empirically validate the effectiveness of our method on real-world time series forecasting benchmarks, showing consistent improvements in both predictive accuracy and covariate alignment over strong baselines.

## 2 RELATED WORK

### 2.1 DIFFUSION MODELS FOR TIME SERIES FORECASTING

Diffusion models have recently been extended to time series forecasting, showing competitive performance in modeling uncertainty and complex temporal dependencies. Diffusion Forecasting Yuan & Qiao (2024) adapts score-based generative models for univariate physical systems, while TimeGrad Rasul et al. (2021) applies denoising diffusion to multivariate time series with temporal masking. More recent works, such as CSDI Tashiro et al. (2021), SSSD Lopez Alcaraz & Strodthoff (2023), TMDMLi et al. and MTSCIZhou et al. (2024a), leverage conditional diffusion for imputation and forecasting tasks, incorporating known time-dependent inputs or partially observed sequences. However, these methods primarily focus on marginal prediction accuracy or data completion and do not explicitly model the semantic alignment between generated samples and future covariates. The sampling process in these models lacks mechanisms to assess or correct semantic consistency with conditioning variables during inference. Our work addresses this gap by introducing a semantic feedback mechanism to guide the diffusion trajectory toward covariate-consistent outcomes.

## 2.2 CONDITIONAL GENERATION AND POSTERIOR MISMATCH

Conditional generative models often suffer from posterior mismatch, where the learned conditional distribution deviates from the true posterior, especially under high-dimensional or multimodal conditioning inputs. This issue, related to the broader challenge of posterior collapse in VAEs Fortuin et al. or mode averaging in diffusion models Yang et al. (2025), can lead to outputs that ignore or underutilize conditioning signals. In time series forecasting, this mismatch manifests as semantic inconsistency between generated trajectories and external covariates—an issue that is rarely evaluated explicitly in prior work. Our approach tackles this problem directly by introducing an inference-time correction procedure based on proxy likelihoods, without modifying the base model's training objective.

## 2.3 IMPORTANCE SAMPLING AND SEQUENTIAL INFERENCE

Importance sampling is a classical technique for approximating complex posteriors, and has inspired a range of sequential Monte Carlo (SMC) methods, such as particle filtering Crisan & Doucet (2002), which maintain a set of weighted hypotheses over time. Recent generative models have incorporated SMC-inspired ideas into sequential latent variable inference Jiang et al. (2023), diffusion-based trajectory control, or policy refinemen. Our work draws inspiration from this line of research by introducing a stepwise importance reweighting mechanism into the diffusion sampling process. However, unlike classical SMC, we do not rely on transition densities or explicit likelihoods—instead, we use a learned semantic consistency score as a proxy likelihood to perform nonparametric posterior correction in the diffusion trajectory space. To the best of our knowledge, we are the first to integrate semantic scoring-based importance weighting into conditional diffusion models for time series forecasting.

## 3 METHOD

This section describes our proposed method, SemGuide, which enhances the semantic consistency of conditional diffusion models for time series forecasting by introducing a semantic scoring mechanism and a stepwise reweighting scheme. The overall framework is illustrated in Figure 1. We begin by formalizing the problem setup, then present the design and training of the scoring network, followed by our inference-time sampling refinement strategy.

## 3.1 PROBLEM SETUP

Let $x_{1:T} = \{x_1, x_2, \ldots, x_T\}$ denote the observed historical time series, and $y_{T+1:T'} = \{y_{T+1}, \ldots, y_{T'}\}$ denote known future covariates, such as planned interventions or external signals. The goal is to forecast the future target sequence $x_{T+1:T'}$, denoted as $x$, conditioned on the covariates $y$.

Diffusion-based time series models approach this problem by learning a conditional generative model $q(x \mid y)$, defined through a denoising diffusion process that transforms noise into data over a sequence of steps. The forward (noising) process gradually adds Gaussian noise to the clean data:

$$q(x_t \mid x_0) = \mathcal{N}(x_t; \sqrt{\alpha_t}x_0, (1 - \alpha_t)I),$$

where $x_0$ is the clean future sequence, $x_t$ is the noised version at step $t$, and $\alpha_t \in (0, 1)$ is the variance schedule. The reverse process is parameterized by a neural network that predicts either the noise or the clean data from $x_t$, yielding the sampling chain:

$$x_{t-1} \sim p_\theta(x_{t-1} \mid x_t, y).$$

These models are typically trained to minimize a denoising objective (e.g., noise prediction loss) using known data and covariates. As a result, the model learns to approximate samples from $q(x \mid y)$, which may indeed match the marginal distribution of valid future trajectories.

However, a critical limitation emerges at inference time. Despite being well-trained, the model may generate samples that, while plausible under the data distribution, are semantically inconsistent

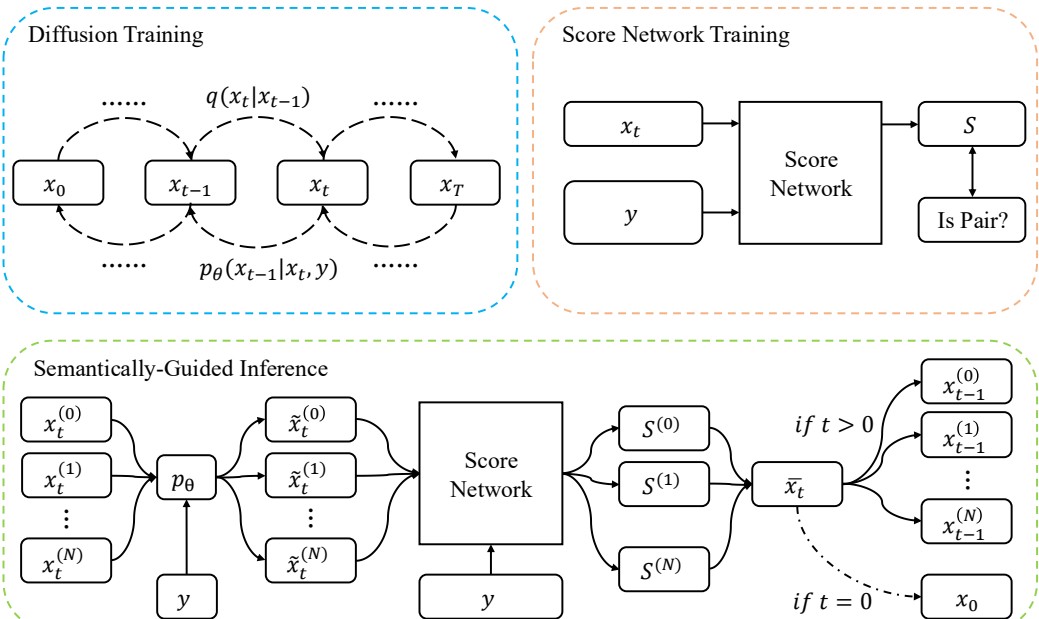

Figure 1: Overview of the SemGuide framework.SemGuide consists of three decoupled components: (1) training a conditional diffusion model using standard denoising objectives; (2) independently training a semantic score network on covariate-aligned trajectory pairs; and (3) applying semantically guided inference to refine the sampling trajectory at test time.

with the conditioning covariates $y$. This is because the diffusion sampling procedure is stochastic, and lacks any mechanism to ensure that intermediate or final predictions remain aligned with the intended semantic meaning of the covariates.

In practice, this inconsistency is typically handled by sampling multiple trajectories (e.g., 50–100) and selecting the pointwise median as the final prediction. While this approach reduces variance and discards extreme samples, it does not fundamentally correct the semantic misalignment problem—it merely lowers the chance of selecting inconsistent outputs, and may blur sharp but consistent forecasts. The generation process should not only denoise the signal but also remain semantically grounded in the covariates throughout the trajectory. To this end, we introduce a semantic scoring mechanism that evaluates the alignment between intermediate samples $x_t$ and the covariates $y$, and use these scores to guide the sampling path via stepwise importance weighting.

### 3.2 SEMANTIC SCORE NETWORK

To evaluate the alignment between partially generated samples and conditioning covariates, we introduce a semantic score network $S(x_t, y) \in [0, 1]$, which estimates a proxy likelihood of the covariates $y$ given an intermediate sample $x_t$. This score will later be used to guide the sampling process toward semantically consistent trajectories.

The score network is trained independently using the original training data, without requiring any additional labels. Each training pair consists of a future ground-truth sequence $x_0$ and its corresponding covariates $y$. To simulate the intermediate states encountered during diffusion sampling, we generate noisy versions of $x_0$ using the same forward noising process employed by the diffusion model. Specifically, for a randomly chosen diffusion timestep $t$, we sample:

$$x_t \sim q(x_t \mid x_0) = \mathcal{N}(\sqrt{\alpha_t}x_0, (1 - \alpha_t)I),$$

where $\alpha_t$ is the predefined noise schedule. The pair $(x_t, y)$ is then treated as a positive example, reflecting a realistic intermediate state that is expected to evolve toward a sample consistent with $y$ under the true reverse process.

To construct negative examples, we randomly sample an alternative future sequence $x_0' \neq x_0$ from the training set, apply the same noising process to obtain $x_t' \sim q(x_t \mid x_0')$, and pair it with the original covariates $y$. The resulting pair $(x_t', y)$ forms a negative example, representing an inconsistent or mismatched trajectory.

Using these constructed pairs, we train the score network as a binary classifier with binary cross-entropy loss, assigning label 1 to positive pairs and 0 to negative pairs. This training encourages the network to assign high scores to semantically compatible $(x_t, y)$ pairs, and low scores to mismatched ones. Because the noise-added states are directly derived from the forward diffusion process, the network naturally learns to operate on the same type of intermediate representations that will be encountered during inference.

Notably, this scoring mechanism is trained entirely offline and remains fixed during inference. It introduces no modification to the original diffusion model and does not require gradient-based updates during sampling. As such, it can be seamlessly integrated with any pre-trained conditional diffusion model to enable semantic consistency assessment at each step of the sampling trajectory.

### 3.3 Semantic-Guided Sampling Correction

At inference time, conditional diffusion models generate samples by iteratively denoising a sequence of latent variables starting from pure Gaussian noise. Given an initial set of particles $\{x_T^{(i)}\}_{i=1}^N \sim \mathcal{N}(0, I)$, the reverse diffusion proceeds in steps from $t = T$ to $t = 1$, refining the latent states toward the final prediction $x_0$.

At each reverse step $t$, we first compute the model-predicted noise for each particle $x_t^{(i)}$ using the conditional diffusion model:

$$\epsilon^{(i)} = \epsilon_\theta(x_t^{(i)}, y, t), \quad \text{for } i = 1, \ldots, N.$$

Next, we compute the preliminary denoised prediction for the previous step $t-1$, using the standard DDPM update formula:

$$\tilde{x}_{t-1}^{(i)} = \frac{1}{\sqrt{\alpha_t}} \left( x_t^{(i)} - \frac{1 - \alpha_t}{\sqrt{1 - \bar{\alpha}_t}} \epsilon^{(i)} \right).$$

We then evaluate each candidate prediction $\tilde{x}_{t-1}^{(i)}$ using the semantic score network, obtaining the alignment score with the conditioning covariates:

$$S(\tilde{x}_{t-1}^{(i)}, y) \approx \tilde{p}(y \mid \tilde{x}_{t-1}^{(i)}),$$

which serves as a proxy likelihood indicating the semantic plausibility of the predicted sample under condition $y$.

These scores are normalized to form importance weights:

$$w_t^{(i)} = \frac{S(\tilde{x}_{t-1}^{(i)}, y)}{\sum_{j=1}^N S(\tilde{x}_{t-1}^{(j)}, y)},$$

which define a discrete distribution over the candidate denoised particles. We then compute the semantic-weighted center of the predicted states as:

$$\bar{x}_{t-1} = \sum_{i=1}^N w_t^{(i)} \tilde{x}_{t-1}^{(i)}.$$

To allow for stochastic exploration around this high-consistency center, we inject noise scaled by the corresponding variance schedule $\sigma_t$ to obtain the next set of particles:

$$x_{t-1}^{(i)} = \bar{x}_{t-1} + \sigma_t z^{(i)}, \quad z^{(i)} \sim \mathcal{N}(0, I).$$

This step ensures that while the sampling process is guided by semantic consistency, it retains the diversity and uncertainty modeling inherent to diffusion models.

---

**Algorithm 1** Semantically-Guided Inference for Conditional Diffusion (SemGuide)

---

**Require:** Conditional diffusion model $\epsilon_\theta$, semantic score network $S(\cdot, \cdot)$, covariates $y$, number of particles $N$, total steps $T$
**Ensure:** Generated prediction $x_0$

1: Initialize particles $\{x_T^{(i)}\}_{i=1}^N \sim \mathcal{N}(0, I)$
2: **for** $t = T$ to 1 **do**
3:     **for** $i = 1$ to $N$ **do**
4:         Predict noise: $\epsilon^{(i)} \leftarrow \epsilon_\theta(x_t^{(i)}, y, t)$
5:         Compute denoised estimate:

$$\tilde{x}_{t-1}^{(i)} \leftarrow \frac{1}{\sqrt{\alpha_t}} \left( x_t^{(i)} - \frac{1 - \alpha_t}{\sqrt{1 - \bar{\alpha}_t}} \epsilon^{(i)} \right)$$

6:         Compute semantic score: $s^{(i)} \leftarrow S(\tilde{x}_{t-1}^{(i)}, y)$
7:     Normalize scores: $w^{(i)} \leftarrow \frac{s^{(i)}}{\sum_{j=1}^N s^{(j)}}$ for $i = 1, \ldots, N$
8:     Compute weighted center: $\bar{x}_{t-1} \leftarrow \sum_{i=1}^N w^{(i)} \tilde{x}_{t-1}^{(i)}$
9:     **for** $i = 1$ to $N$ **do**
10:       Sample $z^{(i)} \sim \mathcal{N}(0, I)$
11:       Update particle:

$$x_{t-1}^{(i)} \leftarrow \bar{x}_{t-1} + \sigma_t z^{(i)}$$

12: **return** $\bar{x}_0$

---

The above procedure is repeated for each timestep $t$ in reverse order, progressively refining the trajectory toward semantically aligned samples. Only the initial step $x_T^{(i)} \sim \mathcal{N}(0, I)$ is purely sampled; all subsequent sets $\{x_t^{(i)}\}_{i=1}^N$ are generated through this semantic-guided reweighting and resampling mechanism.

This process is summarized in Algorithm 1. It can be viewed as a particle-filtering-inspired correction within the latent space of diffusion models, where the learned scoring function provides a non-parametric, model-agnostic proxy likelihood for evaluating and guiding intermediate predictions. Unlike classical filtering, this approach does not require access to explicit likelihoods or transition kernels, making it efficient and easy to integrate with any conditional diffusion backbone.

## 4 EXPERIMENTS

To evaluate the effectiveness of our proposed SemGuide framework, we conduct extensive experiments on real-world time series forecasting tasks. Our goal is to assess both the predictive accuracy and the semantic consistency of generated forecasts under complex conditional structures.

While our method is designed to enhance generative diffusion models, we also include comparisons with a selection of deterministic forecasting methods that directly regress the target sequence without modeling distributional uncertainty. These models, though fundamentally different in formulation, serve as strong baselines for point prediction performance and help contextualize the gains achieved by diffusion-based approaches.

In addition, we compare SemGuide with a range of state-of-the-art diffusion-based forecasters, and apply our semantic guidance mechanism to each, demonstrating its general applicability and effectiveness across different generative backbones. Our evaluation includes both quantitative metrics—covering accuracy and consistency—as well as qualitative analyses of sample behaviors and alignment with conditioning covariates.

Table 1: Forecasting performance on EPF datasets. The upper block reports deterministic models, while the lower block shows diffusion-based generative models. "+Cov." denotes conditioning on future covariates during training; "+SemGuide" indicates the use of our semantic-guided inference at test time. Bold numbers highlight the best overall results, and underlined values indicate improvements brought by SemGuide over the corresponding base model.

| Method | BE | | DE | | FR | | NP | | PJM | |
|---|---|---|---|---|---|---|---|---|---|---|
| Metric | MSE | MAE | MSE | MAE | MSE | MAE | MSE | MAE | MSE | MAE |
| TimeXer | 0.368 | **0.231** | 0.283 | 0.322 | 0.364 | 0.193 | 0.175 | 0.225 | 0.080 | 0.175 |
| TiDE | 0.447 | 0.301 | 0.515 | 0.462 | 0.407 | 0.251 | 0.296 | 0.309 | 0.105 | 0.213 |
| TSMixer | **0.316** | 0.241 | 0.250 | 0.322 | 0.396 | 0.214 | 0.187 | 0.253 | 0.078 | 0.174 |
| TFT | 0.352 | 0.240 | 0.262 | 0.326 | 0.371 | 0.182 | 0.193 | 0.237 | 0.079 | 0.174 |
| iTransformer | 0.339 | 0.246 | 0.258 | 0.327 | 0.378 | 0.205 | 0.203 | 0.250 | 0.077 | 0.171 |
| CSDI+Cov. | 0.347 | 0.265 | 0.257 | 0.299 | 0.262 | 0.174 | 0.135 | 0.218 | 0.057 | 0.131 |
| CSDI+SemGuide | _0.339_ | _0.253_ | _0.241_ | _0.286_ | _0.258_ | _0.169_ | _0.130_ | _0.196_ | _0.053_ | _0.129_ |
| SSSD+Cov. | 0.388 | 0.297 | 0.276 | 0.289 | 0.292 | 0.193 | 0.214 | 0.276 | 0.054 | 0.134 |
| SSSD+SemGuide | _0.381_ | _0.290_ | _0.268_ | **0.281** | _0.279_ | _0.185_ | _0.205_ | _0.269_ | _0.053_ | _0.132_ |
| TMDM+Cov. | 0.402 | 0.266 | 0.348 | 0.359 | 0.333 | 0.236 | 0.247 | 0.255 | 0.091 | 0.169 |
| TMDM+SemGuide | _0.398_ | _0.263_ | _0.346_ | _0.355_ | _0.318_ | _0.228_ | _0.240_ | _0.249_ | _0.083_ | _0.162_ |
| MTSCI+Cov. | 0.397 | 0.292 | 0.245 | 0.295 | 0.257 | 0.168 | 0.131 | 0.182 | 0.055 | 0.130 |
| MTSCI+SemGuide | _0.385_ | _0.283_ | **0.240** | _0.287_ | **0.254** | **0.164** | **0.120** | **0.173** | **0.048** | **0.125** |

## 4.1 EXPERIMENTAL SETUP

### 4.1.1 DATASETS

We evaluate our method on the Electricity Price Forecasting (EPF) datasetLago et al. (2021), a real-world benchmark widely used for high-resolution, covariate-informed forecasting. The dataset comprises day-ahead electricity price data from five major power markets: Belgium (BE), Germany (DE), France (FR), Nord Pool (NP), and PJM (US). Each market provides hourly electricity prices over a six-year period. Following standard practice, we treat the hourly day-ahead electricity price as the target variable, and use publicly available day-ahead load forecasts and generation forecasts as known future covariates. This setting reflects realistic market conditions in which external variables are partially known in advance, and the model must produce price forecasts that are semantically consistent with these future signals.

### 4.1.2 BASELINES

We compare SemGuide against two categories of baselines:

Deterministic forecasting models: TimeXer Wang et al. (2024), TiDE Das et al. (2023), TSMixer Chen et al. (2023), TFT Lim et al. (2021), and iTransformer Liu et al.. These models directly regress future values and do not explicitly model uncertainty.

Diffusion-based generative models:CSDI Tashiro et al. (2021), SSSD Lopez Alcaraz & Strodthoff (2023), TMDM Li et al., and MTSCI Zhou et al. (2024a). These models produce probabilistic forecasts by sampling from learned distributions.

Since most existing diffusion-based methods do not natively support future covariates, we extend them by concatenating future covariates with historical sequences as conditioning input during both training and inference, ensuring a fair comparison under the same information setup. We conduct experiments on 4 * NVIDIA RTX 4090 GPUs. The diffusion models are trained for 500 epochs using the AdamW optimizer with a learning rate of $1e^{-4}$ and weight decay of $1e^{-5}$. The semantic score network is trained for 400 epochs with a learning rate of $1e^{-3}$.

## 4.2 MAIN RESULTS

We summarize the quantitative forecasting results across five electricity markets in Table 1. Among the deterministic baselines, TSMixer and TimeXer achieve competitive performance. However,

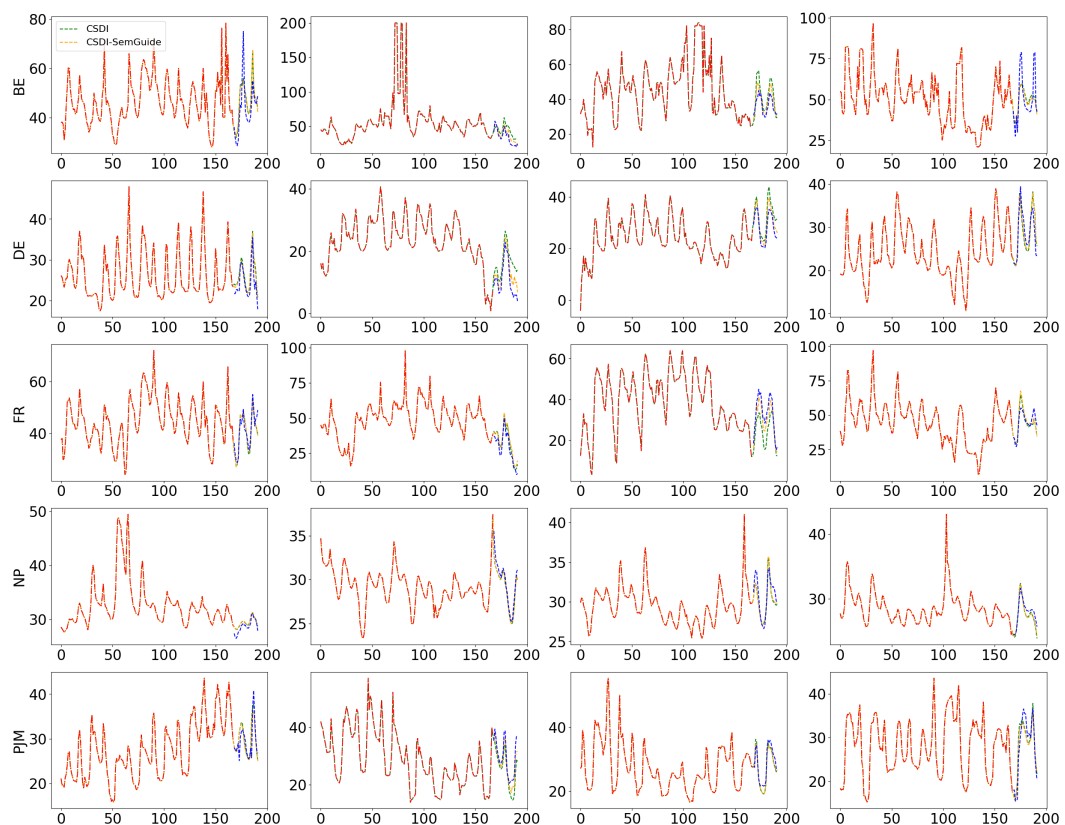

Figure 2: Qualitative comparison of forecast trajectories on selected cases from the EPF dataset. The red dashed line indicates the historical time series, the blue dashed line represents the ground truth, the green dashed line shows the prediction from CSDI, and the orange dashed line denotes the prediction from CSDI-SemGuide.

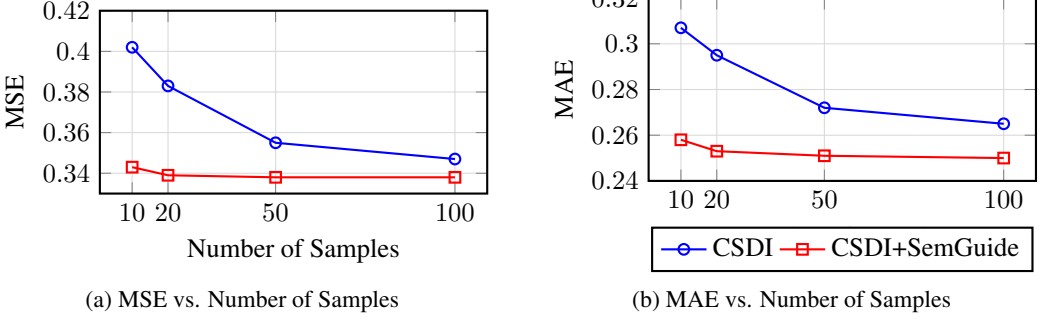

(a) MSE vs. Number of Samples

(b) MAE vs. Number of Samples

Figure 3: Sample Efficiency Analysis. CSDI+SemGuide achieves superior or comparable forecasting accuracy with far fewer particles, indicating improved sampling efficiency.

overall, diffusion-based generative models consistently outperform deterministic methods in terms of both MSE and MAE, particularly under higher uncertainty and more volatile regimes.

Within the generative models, we compare each base method (with 100-sample pointwise median prediction) against its corresponding SemGuide-enhanced version. Across all five markets, SemGuide consistently improves the predictive accuracy of its base models. For instance, CSDI+SemGuide achieves the lowest MSE and MAE on DE, FR, NP, and PJM, outperforming both the base CSDI+Cov. and all other baselines. On the BE market, CSDI+SemGuide also shows improvement over CSDI+Cov. in both metrics. Similar trends are observed for other diffusion

backbones. SSSD+SemGuide reduces both MSE and MAE across all markets compared to its base version. TMDM+SemGuide and MTSCI+SemGuide also consistently outperform their corresponding base models, highlighting the generality of our semantic guidance mechanism.

These results validate the core intuition behind SemGuide: by reweighting particles based on semantic consistency at each sampling step, the model produces samples that not only better align with the conditioning covariates but also yield more accurate point forecasts. In summary, SemGuide is able to deliver both distributional coherence and improved predictive quality across a variety of generative backbones and forecasting settings.

### 4.3 QUALITATIVE ANALYSIS

To better understand how semantic guidance influences the sampling behavior of diffusion models, we visualize forecast trajectories produced by different methods on selected cases from the EPF dataset. Figure 2 shows representative examples comparing the ground truth, median-based predictions from standard diffusion sampling, and SemGuide-enhanced forecasts.

From the selected cases, we observe that while both methods produce reasonable trajectories, forecasts generated by CSDI+SemGuide more closely follow the ground truth patterns, particularly in regions with sharp fluctuations or rapid trend reversals. These visual results suggest that semantic guidance helps the model better capture the temporal structure implied by the conditioning context, even though covariates are not explicitly shown in the figure. The improved trend alignment of SemGuide-enhanced forecasts highlights its effectiveness in enforcing semantic consistency during the sampling process.

### 4.4 SAMPLE EFFICIENCY ANALYSIS

To further evaluate the efficiency of semantic guidance, we conduct an experiment comparing two sampling strategies based on the CSDI model: (1) the standard approach of generating multiple samples and taking the pointwise median, and (2) our SemGuide approach using stepwise reweighting with a reduced number of particles. Specifically, we vary the number of samples for the median-based baseline and the number of particles for SemGuide, and report the resulting MSE and MAE across all five datasets.

The results, summarized in Figure 3, show that while the baseline method requires at least 100 samples to achieve stable predictive performance, SemGuide reaches comparable or even better accuracy and consistency with as few as 10 to 20 particles. This demonstrates that semantic guidance not only improves prediction quality but also leads to significantly more efficient sampling, reducing computational cost without sacrificing accuracy.

These findings reinforce our core hypothesis: reweighting intermediate diffusion states based on semantic consistency provides a stronger signal than naïve aggregation across unstructured samples. By explicitly promoting alignment with covariates during sampling, SemGuide enables high-quality predictions with far fewer forward passes through the model.

## 5 CONCLUSION

In this work, we propose SemGuide, a novel inference-time framework for enhancing semantic consistency in conditional diffusion models. Our method introduces a lightweight scoring network to evaluate the alignment between intermediate diffusion states and covariates, and incorporates a stepwise reweighting strategy to guide the sampling process toward semantically faithful trajectories.

Unlike existing approaches that rely on large-scale sampling and heuristic aggregation, SemGuide provides an explicit semantic correction mechanism without modifying the base diffusion model or its training objective. It is compatible with a wide range of generative backbones and can be seamlessly applied in practical settings where covariate-aware prediction is essential.

Extensive experiments on real-world electricity price forecasting tasks demonstrate that SemGuide significantly improves both pointwise predictive accuracy and semantic consistency, outperforming strong deterministic and generative baselines. Our method achieves high-quality forecasts with fewer particles, offering an efficient and controllable alternative to traditional sampling strategies.

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
