# OpenReview forum: "Semantically-Guided Inference for Conditional Diffusion Models: Enhancing Covariate Consistency in Time Series Forecasting"
_ICLR.cc/2026/Conference — ICLR 2026 Conference Withdrawn Submission_

### Official Review · Reviewer_b5UE · 2025-10-17

**Soundness:** 3
**Presentation:** 3
**Contribution:** 2
**Rating:** 2
**Confidence:** 3

**Summary:**

This paper proposes SemGuide, a lightweight inference-time framework that enhances the semantic alignment between generated forecasts and conditioning covariates in diffusion-based time series models. By introducing a semantic scoring network and stepwise importance reweighting, SemGuide guides sampling toward covariate-consistent trajectories without modifying model training. Experiments on electricity price forecasting show improved accuracy, consistency, and sampling efficiency across various diffusion backbones.

In Chapter 1, the authors state “We identify and address a core limitation in conditional diffusion models,” but they do not explain or demonstrate why diffusion models inherently have this problem.

I do not clearly see the motivation of this work—why introducing a semantic score network can improve model performance, or where the essential information gain comes from.

The motivation of the paper is unproven and not clearly explained, leaving it unclear whether it arises from an observed phenomenon or theoretical reasoning.

The experiments are far from sufficient; using only MSE and MAE metrics is inadequate.

Furthermore, in the limited experiments presented, the performance improvement appears quite marginal.

Finally, what is the actual connection between this work and time series forecasting? Does the proposed model address a problem of diffusion models or a problem of time series forecasting itself?

**Strengths:**

see summary

**Weaknesses:**

see summary

**Questions:**

see summary

---

### Official Review · Reviewer_5rGk · 2025-10-26

**Soundness:** 1
**Presentation:** 2
**Contribution:** 1
**Rating:** 2
**Confidence:** 5

**Summary:**

This paper proposes SemGuide, an inference-time guidance mechanism for conditional diffusion models that combines score matching with particle filtering-inspired importance reweighting to enhance covariate consistency in time series forecasting. The application to covariate-conditioned time series forecasting is particularly interesting, as it addresses scenarios where future-known covariates, like control signals or external features, should semantically constrain generated trajectories. Experimental results are shown for 5 datasets, all from the electricity price forecasting domain.

Primarily, the term "semantic guidance" is loosely used and never rigorously defined. While this terminology may be appropriate in language domains, its application to time series is questionable; terms like "covariate consistency" makes sense and it would have been meaningful to use that throughout. Compounding this issue, the paper provides no direct quantitative metrics for measuring semantic consistency, relying instead on standard forecasting metrics (MSE/MAE) that do not specifically evaluate covariate alignment.

The qualitative analysis in Figure 2 is unconvincing. The visual comparison between median-based predictions and SemGuide-enhanced forecasts does not clearly demonstrate superiority, with both approaches appearing similarly aligned with ground truth. The table shows marginal improvement, but one wonders why despite being a probabilistic method, confidence intervals and statistical significance analysis is missing.

Overall, the paper does identify an interesting and unexplored area, but overall contributions, including experimental validation (or theoretical justification), are inadequate for ICLR community standards.

**Strengths:**

1. Identifying a real, understudied problem in conditional diffusion models involving future covariate conditioned time series forecasting

2. Attempting to propose model-agnostic solution that doesn't require retraining of the backbone diffusion models

3. In limited experiments, sampling efficiency is shown (though there are issues as listed below)

4. Figure 1 is well-made, and the algorithms are clearly explained. The experiment setup is also well written.

**Weaknesses:**

1. The first major weakness is the weak experimental support for the major claims. The method is only tested on electricity price forecasting (EPF) dataset. No evaluation on other time series domains (weather, traffic, healthcare, finance) used in the TSF community is shown to demonstrate generalizability. Forecast length of hourly price for day ahead (24) is only used, severely limiting the demonstration of the claims.

2. Despite the core contribution being semantic alignment, there's no quantitative metric that directly measures covariate-forecast consistency.

3. Despite being a probabilistic method, there is no use of metrics commonly used for probabilistic time series forecasting. Even for the point estimates, rigorous uncertainty quantification and statistical significance analysis is missing.

4. Other guidance-based diffusion methods must be included in the baseline, and the lift due to the inference time guidance must be demonstrated in an expanded experimental evaluation.

5. No strong theoretical analysis of why this reweighting scheme should improve semantic consistency. Firstly, semantic consistency itself is not defined (see above). No analysis of whether the method using the importance weight reported in the paper converges to the desired distribution (is there even a desired distribution?)

6. Empirical evidence of score quality and semantic consistency is not provided.

7. Figure 2 shows forecast comparisons but does not clearly demonstrate the semantic consistency improvement. The improvement over the baseline with no SemGuide seems marginal or non-existent in some cases (I did zoom in a lot to see across the provided examples). I do agree that in some cases there is a difference, but not something to write home about.

8. For the score network, the architecture is not provided. Also, binary classification with random negative sampling seems preliminary. The selected random negatives may not cover the actual failure modes, right? What are the failure modes?

9. Details of the covariate structure is missing in the experiment section.

10. Computational overhead of inference time reweighting by SemGuide is not provided and compared against median method.

Minor matter: Please use citep. It was quite uncomfortable to read the sentences with the author-year references that start immediately after the close of the sentence or together with the word.

**Questions:**

1. Could you provide a rigorous, quantitative definition of "semantic consistency" or "semantic alignment" in the context of time series forecasting? More importantly, why not introduce direct metrics that specifically measure covariate-forecast consistency (e.g., correlation between covariate patterns and forecast patterns, conditional distribution distances, or task-specific consistency scores)? Without such metrics, how can readers verify that your method actually improves semantic alignment rather than simply improving general forecast accuracy through a different aggregation mechanism?

2. Could you provide comprehensive ablation studies examining: (a) the impact of negative sampling strategies (random vs. hard negatives vs. temporally misaligned), (b) the effect of noise levels during score network training, (c) sensitivity to the number of particles N, (d) which diffusion timesteps benefit most from guidance, (e) the contribution of importance reweighting vs. simply selecting the highest-scoring particle, and (f) computational cost comparisons (wall-clock time, memory usage) between your method and baseline median aggregation? These analyses would clarify which design choices are essential versus incidental and help practitioners adapt your method to their settings.

3. Could you provide explicit comparisons (conceptual and empirical) with established guidance approaches (from image and language) adapted to TSF? What prevents using existing methods (say classifier guidance, classifier-free guidance etc), and what specific advantages does your score-matching with particle-filtering approach offer?

4. I also have many questions on modifying diffusion based methods for covariate conditioned forecasting. But, I will only list two as I understand that your focus is on not modifying the backbone: (i) Is extending the baselines by concatenating future covariates the optimal use?; (ii) Could alternative conditioning mechanisms (cross-attention, FiLM layers, or other conditioning architectures) yield stronger baseline performance?

5. All experiments are conducted on electricity price forecasting with specific types of covariates (load and generation forecasts). Could you demonstrate the method's effectiveness on diverse time series domains (e.g., weather forecasting with meteorological covariates, traffic prediction with planned events, healthcare monitoring with intervention schedules, or financial forecasting with policy indicators) and different types and number of covariates to establish generalizability? What characteristics of the electricity domain might make it particularly amenable to your approach, and under what conditions might the method fail or require adaptation?

---

### Official Review · Reviewer_Zh3P · 2025-11-01

**Soundness:** 1
**Presentation:** 1
**Contribution:** 2
**Rating:** 2
**Confidence:** 5

**Summary:**

The paper introduces SemGuide, which uses a "Score Network" to predict alignment of current trajectories with future covariates. These scores are then used to compute a weighted average across the candidates. Empirically, this improves the forecasts of time series diffusion models in terms of MAE and MSE.

**Strengths:**

- The results improve consistently on the MSE and MAE.
- The idea of using a score network to distinguish which samples are reasonable and which are not is interesting.

**Weaknesses:**

- There is no background section. Diffusion models are introduced in their methodology section without citing the corresponding papers. These works should be cited accordingly and not only in the introduction.
- The related work section is missing many important references from the diffusion, conditional generation, and time series domains.
- The notation is inconsistent and makes it hard to follow the method section. The subscript is used to describe both the diffusion time and the temporal dimension (e.g., L142 and L150).
- Experiments do not back up the claim that conditional diffusion is semantically inconsistent. Conditional diffusion models have previously demonstrated competitive performance in forecasting and imputation tasks, showing the ability to generate semantically consistent samples.
- The core idea of averaging trajectories is not too convincing. The generated samples are merged with a weighted average after each diffusion step, removing any variation from the generated trajectories. It would be interesting to see confidence intervals in Figure 2.
- The evaluation is performed using MSE and MAE, which are typically metrics used to evaluate point forecasters. A probabilistic metric, e.g., CRPS, would be more appropriate in this case.
- Runtime comparison and standard deviations are missing.

Formatting and other:

- Incorrect use of textual and parenthetical citations.
- L46: copy/paste error or typographical errors?

**Questions:**

See weaknesses and:

- How many samples do you use to evaluate SemGuide? Since you have multiple samples in each intermediate step, I would assume that you only use one generated sample in the end. Can you confirm this?
- How does the runtime compare to the baselines?

---

### Official Review · Reviewer_6LMG · 2025-11-10

**Soundness:** 1
**Presentation:** 2
**Contribution:** 2
**Rating:** 2
**Confidence:** 3

**Summary:**

This paper proposes SemGuide, a plug-and-play inference-time method to improve semantic consistency between generated forecasts and conditioning covariates in diffusion-based time series forecasting models. The key innovation is a semantic scoring network trained to assess alignment between intermediate diffusion states and future covariates. During inference, the method uses stepwise importance reweighting of multiple particles based on these scores to guide sampling toward more covariate-consistent predictions. The approach is model-agnostic and evaluated on electricity price forecasting tasks.

**Strengths:**

**Non-Invasive**: It operates only at inference time, requiring no retraining or architectural changes to the base model.

**Model-agnostic design**: The plug-and-play nature means SemGuide can be applied to any pre-trained conditional diffusion model without retraining.

**Weaknesses:**

**Reliance on Score Network Design**: The effectiveness of the scoring network, which is trained on positive and negative pairs, may be highly sensitive to the quality and diversity of these pairs. The paper does not analyze the robustness of the method to different design choices or potential limitations of the score network.

**Limited Empirical Validation**: The empirical evaluation is restricted to a single domain (electricity price forecasting), albeit across five markets. The work lacks validation on other time series domains such as healthcare, traffic, or weather, where covariate semantics may differ significantly.

**Inconsistency Between Abstract and Experiments**: The abstract mentions multimodality as a feature, but this claim is not supported by any experiments or results presented in the paper. I suggest removing it.

**Lack of Discussion on Computational Overhead**: While the method is described as being efficient in terms of sample count, the computational cost of running the scoring network for every particle at each denoising step is not discussed or analyzed. While the authors describe the network as "lightweight", they do not provide any details on the scoring network's architecture or parameter count.

**Insufficient Semantic Metrics**: The paper claims to improve "semantic alignment" but only reports standard metrics like Mean Squared Error (MSE) and Mean Absolute Error (MAE). No direct metrics are provided to quantify covariate consistency, such as the correlation between forecast features and covariate patterns.

**Lack of a detailed ablation on the score network**: The paper lacks a thorough ablation study to understand the score network itself. For example, the impact of alternative network architectures and fewer/no negative samples. Hence, it remains uncertain whether the score network is truly essential.

**Presentation weakness**: Most of citations use incorrect parenthesis placement: "TimeXer Wang et al. (2024)" → should be "TimeXer (Wang et al., 2024)". This formatting error appears consistently throughout the paper, suggesting systematic misuse of \cite{}, \citep{} in LaTeX.

In summary, the primary weakness of this paper is **soundness**, i. e. the lack of sufficient empirical evidence to fully support its central claims. Key areas that would strengthen the work include expanding the evaluation to diverse domains and conducting thorough ablation studies.

**Questions:**

1. How sensitive is SemGuide's performance to the architecture and training of the semantic score network? For instance, what happens if the score network is poorly calibrated or fails to generalize to the test distribution?
2. Does focusing on high-scoring particles reduce forecast diversity? This could be problematic for uncertainty quantification.

---

### Note · Authors · 2025-11-12

I have read and agree with the venue's withdrawal policy on behalf of myself and my co-authors.